# Factors Affecting Physical Activity in People with Dementia: A Systematic Review and Narrative Synthesis

**DOI:** 10.3390/bs13110913

**Published:** 2023-11-08

**Authors:** Ruth W. Feenstra, Liese J. E. de Bruin, Marieke J. G. van Heuvelen

**Affiliations:** Department of Human Movement Sciences, University of Groningen, University Medical Center Groningen, P.O. Box 196, 9700 AD Groningen, The Netherlands

**Keywords:** exercise, Alzheimer’s disease, cognitive impairment, barrier, facilitator, motivator

## Abstract

Physical activity (PA) has positive effects on the physical and cognitive functioning of people with dementia. Knowledge about what limits and stimulates people with dementia to participate in PA is essential to promote effective PA implementation and enhance PA levels. Previous reviews primarily included opinion-based studies, using data from interviews, focus groups or dyads. By including implementation studies, we aimed to elaborate on previous reviews by identifying new barriers to PA and new facilitators and motivators for PA. We conducted systematic searches in Pubmed, PsychInfo and Web of Science for studies published up to the 21st of September 2021. Search terms were related to the population of people with dementia, PA interventions and implementation outcomes. Studies were included if PA participation was investigated during actual PA implementation. No restrictions were made regarding study design, date of publication, PA type or outcome measures. Studies not implementing PA or not evaluating the implementation were excluded. Based on 13 empirical studies, we identified 35 barriers, 19 facilitators and 12 motivators. Of these, 21 barriers, 11 facilitators and 4 motivators were not identified by previous reviews. New factors are related to the support for people with dementia from informal and formal caregivers, e.g., revealing the importance of a trusting relationship. Furthermore, support for staff from the institution or an external party is needed to overcome doubts about PA, for example, related to safety and effects. New factors also suggested specific recommendations for the content and organization of the PA intervention, for instance, related to how to give instructions. Overall, factors affecting PA identified with opinion-based or implementation studies are complementary. Our extended overview shows the complexity of PA implementation and may help to personalize PA, develop implementation strategies, facilitate actual PA implementation and free up resources needed for effective implementation.

## 1. Introduction

Dementia is a global epidemic with increasing prevalence from 55 million people in 2020 to 78 million in 2030 to 139 million in 2050 [1]. Dementia is an umbrella term for a variety of diseases that are characterized by a substantial global decline in cognitive function from a previous level of functioning that is not attributable to alteration in consciousness [2,3]. Specific cognitive domains that may be affected by dementia are complex attention, executive function, learning and memory, language, perceptual-motor and social cognition [4]. Furthermore, people with dementia often experience physical limitations [5]. These cognitive and physical limitations interfere with everyday activities resulting in disability and dependency [3,6]. Since there is no cure yet, treatment is focused on delaying the onset [7], slowing progression [8] and alleviating symptoms [9,10]. Several pharmacological and non-pharmacological interventions are available [11,12,13,14]. Physical activity (PA) is one such non-pharmacological intervention and has received increased attention in recent years [15].

Many studies have revealed the beneficial effects of PA in people with dementia. PA improves physical functioning, including increased functional mobility [16], gait speed [17], strength, balance and endurance [18]. PA may reduce the risk of falls [19], improve the ability to perform activities of daily living (ADLs) [20,21,22] and promote independent functioning [23]. Next to the physical and functional benefits of PA, there is evidence that PA could alleviate global cognitive decline [24,25,26,27]. Some studies even showed beneficial effects of PA on specific cognitive functions, such as executive function and memory [28,29,30]. These findings are essential for people with dementia as these cognitive functions are most affected by the disease. Even with less strong evidence for specific cognitive functions, PA remains important because of its effects on physical, functional and global cognitive functioning. Despite the beneficial effects of PA, several studies revealed that both institutionalized and community-dwelling people with dementia are sedentary for most of the day and perform less PA compared to healthy peers [31,32,33]. To achieve and maintain the beneficial effects of PA, physical activities should be integrated into people’s daily lives [34,35]. Therefore, knowledge about factors that influence PA participation is essential to promote effective PA implementation and, consequently, to enhance and maintain the PA levels of people with dementia. 

Stubbs et al. [36] previously distinguished several types of factors associated with PA participation. These factors include barriers, which can limit PA participation, motivators, which can motivate PA participation, and facilitators, which can facilitate PA participation. Since PA implementation is a complex process, these factors should be considered on multiple levels. Along with the characteristics of the individuals involved, the intervention itself, the patient’s needs and resources, the characteristics of the organization and the process of implementation should be taken into account [37]. To increase PA participation, we need a broad approach to understand what limits, facilitates or motivates PA participation in people with dementia. 

Two methods can be used to identify barriers to PA, and facilitators and motivators for PA. First, the opinion-based approach acquires information through interviews, focus groups and dyads with people with dementia, their informal caregivers and sometimes health professionals (e.g., nurses, physiotherapists) or exercise providers. For example, a qualitative study interviewed people with dementia and their caregivers to obtain in-depth information on PA beliefs and experiences [38]. Second, the empirical approach implements an actual PA intervention in the daily lives of people with dementia to identify the barriers, facilitators or motivators that affect PA participation. The empirical approach involves intervention studies assessing the acceptability, feasibility and/or adherence of the implemented PA intervention. These studies gather information specific to the implemented PA intervention through, e.g., observations, keeping a log of activities, focus groups and feedback from the involved participants, caregivers, instructors or health professionals. For example, a multicomponent PA intervention gained information on PA participation by evaluating each session with a feasibility questionnaire and by collecting participants’ feedback with focus groups [39]. 

Two prior reviews identified barriers, facilitators and motivators affecting PA in people with dementia. Van Alphen et al. [40] evaluated and summarized primarily opinion-based barriers, facilitators and motivators. Another review by Vseteckova et al. [41] used both opinion-based and empirical methods to examine barriers and facilitators, but limited itself to adherence to walking group exercise. To our knowledge, no previous review has evaluated the barriers, facilitators and motivators affecting PA using empirical data from actual implementations. Therefore, this review aims (1) to identify the barriers to, and facilitators and motivators for, PA in people with dementia, using empirical data from actual PA implementation; (2) to evaluate whether these factors affecting PA are corresponding or complementary to the factors previously identified using an opinion-based approach; and (3) to provide an overview of factors identified by the opinion-based and the empirical approach.

## 2. Materials and Methods

### 2.1. Literature Search

This systematic review followed the Preferred Reporting Items for Systematic Reviews and Meta-Analyses (PRISMA) guidelines, except for the section about meta-analyses [42]. The search was conducted in Pubmed, PsychInfo and Web of Science and included terms related to the population of people with dementia (‘dementia’ or ‘cognitive impairment’ or ‘neurocognitive disorder’), physical activity interventions (‘physical activity’ or ‘exercise’ or ‘sedentary’ or ‘inactivity’) and implementation-related outcomes (‘implementation’ OR ‘barrier’, OR ‘facilitator’ OR ‘motivator)’ (see Appendix B for the full search string by source). The reference lists of the included studies were screened for additional papers. The search was executed by R.W.F. and finished on the 28th of September 2021. The review was not registered.

### 2.2. Inclusion and Exclusion Criteria

Inclusion criteria were the following: PA participation was investigated in an actual implementation of PA intervention in institutionalized and/or community-dwelling people with (mild) dementia, the average age of the participants was over 65 years and factors that limit (barriers), facilitate (facilitators) or motivate (motivators) PA were identified. The latter (factors) was only used for full text screening. No restrictions were made regarding research design, date of publication, PA type or outcome measures. Health or lifestyle studies combining PA with other intervention types were included as well. However, studies that focused only on other interventions, such as dementia care interventions, were excluded. Studies were also excluded if only the effects of the PA intervention were reported. Other exclusion criteria were: not implementing PA (i.e., opinion-based), incomplete or unfinished studies, protocol studies, full text not available and not written in English or Dutch. 

After removing duplicates, two authors (R.W.F. and L.J.E.d.B.) independently screened titles and abstracts for eligibility (agreement 98.3%). In case of disagreement, studies were screened full text. Based on the same criteria, the same authors independently screened the full text of potentially eligible studies identified by title and abstract (agreement 86.3%). Disagreements about the eligibility of a study were then resolved by mutual discussion and discussion with the third author (M.J.G.v.H.). 

### 2.3. Data Extraction and Processing

For each study, we extracted study design, information regarding participants (number, diagnosis, age, gender, use of walking aid and living situation) and information about the PA intervention (type, intervention). Furthermore, we extracted reported factors affecting PA. Studies reported factors directly (e.g., reported in a table or explicitly mentioned as a factor) or indirectly (mentioned in text, not explicitly indicated as a factor). We classified the reported factors as barriers, facilitators and motivators. We distinguished facilitators and motivators based on the corresponding definition: facilitators are factors that make it easier for people with dementia to participate in PA, while motivators directly motivate people with dementia to participate in PA. We extracted information regarding the success of the intervention formulated objectively (e.g., adherence/attendance rate) and subjectively (e.g., conclusion of the author). Finally, we extracted information regarding recommendations given by the authors. The data extraction was performed by one author (R.W.F.).

Next, we categorized the reported barriers, motivators and facilitators in accordance with the six themes of the social-ecological model of McLeroy et al. [43] and in agreement with Van Alphen, Hortobágyi and van Heuvelen [40]. These themes include intrapersonal factors (physical and mental health and preferences), interpersonal factors (support and social identification) and community factors (organization and environment). Within these themes, we ordered the barriers, motivators and facilitators by the number of studies that reported the specific factors.

Finally, we aggregated our implementation-based factors with the previously identified barriers, motivators and facilitators of Van Alphen, Hortobágyi and van Heuvelen [40] and Vseteckova et al. [41]. We also categorized the opinion-based barriers, motivators and facilitators according to the aforementioned description. In the discussion section, we compared factors not identified in previous reviews with related opinion-based barriers, motivators or facilitators. 

### 2.4. Methodological Quality Assessment

Without restrictions on study design, we included qualitative, quantitative and mixed designs. For the methodological quality assessment of studies with a qualitative design, we used the Critical Appraisal Skills Program (CASP) [44]. The CASP is the preferred tool for assessing the methodological quality of qualitative studies in healthcare research and is often used in reviews related to dementia care [45,46]. The CASP qualitative research studies checklist is a frequently recommended tool and consists of ten questions divided in three domains: ‘Are the results of the study valid?’, ‘What are the results?’ and ‘Will the results help locally?’. 

For the methodological quality assessment of studies with a quantitative or a mixed design, we used the Quality Assessment Tool for Quantitative studies developed by the Effective Public Health Practice Project (EPHPP) [47]. The EPHPP has been validated for use in public health research, was previously used in dementia-related research and is suitable for multiple (combined) study designs [48,49]. The EPHPP consists of six domains: selection bias, study design, confounders, blinding, data collection method and withdrawals and drop-out. One author (R.W.F.) assessed the methodological quality of included studies via full text screening. Uncertainties were settled by discussion with another author (M.J.G.v.H.).

## 3. Results

### 3.1. Methodological Quality of Studies

Table 1 shows the results of the quality assessment of the studies with a qualitative design. Included studies generally scored sufficiently on the study design, the method and results of the study. However, we qualified the results section of the study of Donkers et al. [50] as low, since the main themes resulting from the thematic analysis remained unclear. In addition, it was not clear how the authors derived barriers and facilitating factors from the qualitative data.

For all studies, we scored item 6 (has the relationship between researcher and participants been adequately considered?) as unclear (Table 1). The reason is that none of the studies reported about the researchers’ own role, potential bias and influence during the formulation of the research questions or data collection, including sample recruitment and choice of location. In addition, none of the studies reported about how the researcher(s) responded to events during the study and whether they considered the implications of any changes in the research design. Table 2 shows the results of the quality assessment of quantitative and mixed studies. Considering the difficulty of including a representative sample from the population of people with dementia, we qualified the selection bias as sufficient in all included studies (domain A). Domain B concerns the study design, with randomized controlled trials and randomized multiple baseline designs rated as strong, nonrandomized controlled trials as moderate and all other designs as weak. Most studies failed to report on confounders (domain C). None of the studies with a single sample of participants considered confounders. Sondell et al. [51] and Tak et al. [52] included two samples but did not analyze the influence of confounders. None of the studies blinded researchers or participants (domain D), although Dawson et al. [53] mentioned other attempts to minimize this bias. Some studies mentioned the validity and/or reliability of data collection tools, but most studies failed to reflect on the data collection methods (domain E). Since most studies were interested in barriers and facilitators of the PA implementation, withdrawals and drop-outs were extensively reported (domain F). 

**Table 1 behavsci-13-00913-t001:** Quality assessment of qualitative studies (CASP).

Reference	Study Design				Method				Results	
	1	2	3	4	5	6	7	8	9	10
Barrado-Martín et al. [54]	+	+	−	+	+	?	+	+	+	+
Barrado-Martín et al. [55]	+	+	+	+	+	?	+	+	+	+
Donkers, van der Veen, Vernooij-Dassen, Nijhuis-van der Sanden and Graff [50]	+	+	+	−	+	?	+	−	−	−
Hancox et al. [56]	+	+	+	−	+	?	+	+	+	+
MacAndrew et al. [57]	+	+	+	+	+	?	+	−	+	+

+ yes/good; − no/not good; ? cannot tell. 1. Was there a clear statement of the aims of the research? 2. Is a qualitative methodology appropriate? 3. Was the research design appropriate to address the aims of the research? 4. Was the recruitment strategy appropriate to the aims of the research? 5. Were the data collected in a way that addressed the research issue? 6. Has the relationship between researcher and participants been adequately considered? 7. Have ethical issues been taken into consideration? 8. Was the data analysis sufficiently rigorous? 9. Is there a clear statement of findings? 10. Is the research valuable?

**Table 2 behavsci-13-00913-t002:** Quality assessment of the quantitative and mixed studies (EPHPP).

Reference	A	B	C	D	E	F
Dawson, Judge and Gerhart [53]	+	+	+	+/−	+/−	+/−
Henskens et al. [58]	+	+/−	+	−	+	+
Henwood et al. [59]	+/−	+/−	+/−	−	−	+/−
Kruse et al. [60]	+/−	−	−	−	+/−	+/−
Sondell, Rosendahl, Gustafson, Lindelöf and Littbrand [51]	+	+	−	−	−	+
Tak, van Uffelen, Paw, van Mechelen and Hopman-Rock [52]	+	+	−	−	−	+
Teri et al. [61]	+	+	+	−	−	+
Yu and Kolanowski [62]	+/−	−	−	−	+/−	−

+ strong; +/− moderate; − weak. A = selection bias; B = study design; C = confounders; D = blinding; E = data collection method; F = withdrawals and drop-outs.

### 3.2. Study Characteristics

The search strategy yielded 1322 records after removing the duplicates. After assessing the full text eligibility of 73 studies, we included thirteen studies. Figure 1 shows the flowchart of the study selection process. 

Included studies implemented an actual PA intervention into the daily lives of people with dementia and evaluated (the process of) the implementation. All studies aimed to assess either the applicability, feasibility, adherence or delivery process of the PA intervention. Table 3 summarizes the characteristics of the included studies. 

### 3.3. Participant Characteristics

The thirteen studies included a total of 702 people with dementia. The sample size ranged from 2 to 255 participants. 

Two studies included only participants with Alzheimer’s Disease [61,61], one study included participants with Mild Cognitive Impairment (MCI) [52], six studies included participants with different types of dementia [50,51,54,55,56,58] and four studies did not mention the specific diagnosis [53,57,59,60]. Overall, the most prevalent diagnosis was Alzheimer’s Disease (51.9%), followed by MCI (19.3%) and vascular dementia (7.1%). Eight studies reported average MMSE scores, showing various results between 6.5 and 28.4. The average age of participants ranged from 73.9 to 86.6 years. All studies included both male and female participants. Eight studies included community-dwelling or community-residing participants, while five studies included participants from nursing homes. Four studies included a total of 67 caregivers.

### 3.4. Characteristics of PA Interventions

Included studies implemented different types of PA interventions. Six studies implemented a single exercise type, such as Tai Chi, a walking program, aerobic exercise or aquatic exercise. Five studies implemented a multicomponent PA intervention. Other PA interventions were a social fitness program and movement-oriented restorative care. The duration of the PA interventions ranged from 3 to 52 weeks. 

### 3.5. Success of Implementation of PA Interventions

The success of the PA implementation was rated objectively (e.g., adherence/attendance rates; % of recommended PA completed or % attended sessions) and/or subjectively (e.g., conclusion of author); see also Appendix A. Seven studies reported high adherence rates (72–99%) and evaluated their implementation as successful. One study did not report adherence rates, but subjectively reported their implementation as successful. The authors of four studies did not subjectively report on success, but mentioned recommendations for future implementations. 

### 3.6. Barriers, Facilitators and Motivators for Physical Activity

We identified 35 barriers to PA, and 19 facilitators and 12 motivators for PA (see Appendix A for factors derived from included studies and S2 for an overview of empirical identified factors). Fourteen barriers, eight facilitators and eight motivators correspond to previously identified factors using an opinion-based approach. Consequently, 21 barriers, 11 facilitators and 4 motivators were new. Table 4 shows a complete overview of the barriers to PA, and facilitators and motivators for PA identified with implementation studies and previous opinion-based studies. 

## 4. Discussion

### 4.1. Complementary Findings

Based on implementation studies, we identified 35 barriers to PA, 19 facilitators and 12 motivators for PA in people with dementia. These factors partly correspond to previous reviews of Van Alphen, Hortobágyi and van Heuvelen [40] and Vseteckova, Dadova, Gracia, Ryan, Borgstrom, Abington, Gopinath and Pappas [41]. However, the opinion-based approach in previous reviews and the current empirical approach also complement each other. In total, we reported 45 barriers; 14 barriers were identified by both approaches, 10 barriers were only identified by the opinion-based approach and 21 barriers only by our empirical approach. For the 33 facilitators, 8 facilitators correspond, 14 facilitators were only identified in previous studies and 11 facilitators are new. For the 20 motivators, 8 motivators correspond, 8 motivators were only identified by previous reviews and 4 motivators are new. Apparently, the empirical approach was relatively successful at identifying barriers to PA, while the opinion-based approach was relatively successful at identifying facilitators and motivators. In the next part of this discussion, the new factors are discussed for each subheading on the intrapersonal, interpersonal and community level of the socio-ecological model. 

### 4.2. New Findings on the Intrapersonal Level

#### 4.2.1. Physical or Mental Health

The new barriers are ‘low sense of efficacy’ and ‘lack of confidence at home’ for the person with dementia (or caregiver) [55]. In line with the self-efficacy theory of Bandura [68], a low sense of efficacy refers to the person with dementia not believing in their capability to perform the exercises. Consequently, this will hamper PA adherence and maintenance. This new barrier sharpens and extends the previously discovered barrier, ‘belief in the person with dementia being unable to complete exercises’ [34], with an explicit reference to the capabilities of the person with dementia and inclusion of the perspective of the caregiver. Self-efficacy is an important determinant of health behaviour and PA performance [68]. Self-efficacy for PA can be promoted by, e.g., positive experiences, belief in the beneficial effects of PA, enjoyment and positive encouragement of a competent instructor [68,69]. Indeed, participating in a suitable PA intervention can positively influence the self-efficacy of the person with dementia through feelings of empowerment and belonging [69,70,71]. Self-efficacy by proxy of the caregiver also plays an important role. A low level of self-efficacy may lead to overprotective behaviour towards the person with dementia, which may discourage the person with dementia from PA. Increased self-efficacy appears essential to maintain PA [71]. ‘Lack of confidence at home’ experienced by the person with dementia or the caregiver hinders the person’s ability to do exercises, suggesting that more professional support is needed at home. Included studies showed that supportive materials can provide some, but limited, guidance [54,55,56]. Regular home visits or phone contacts with health professionals or instructors, especially for persons with severe cognitive impairments, should be available when needed [56,65]. 

A new facilitator is ‘behavioural problem solving’ [61] which overcomes the previously mentioned barrier ‘disruptive behaviour PwD’ [35,62]. Behavioural problems can be alleviated by educating family caregivers and staff on effective communication with the person with dementia. In addition, increasing pleasant events, such as providing a delicious treat before the PA, can help to overcome behavioural problems [61,72]. On top of these common strategies, individualized non-pharmacological strategies are necessary to address the unique symptoms of people with dementia [73,74]. For example, depending on the person, people with anxiety may need more reassurance or distractions during exercise and may benefit from routine schedules [73].

#### 4.2.2. Individual Preferences

New barriers are ‘negative perception or dislike of PA’ [52,55], ‘boredom or lack of enthusiasm’ [53], ‘not enjoying PA’ [54], ‘lack of accommodating PA’ (to preferences) [52] and ‘difficulty finding appropriate activities’ [50]. The new barriers elaborate on the previously found barrier ‘people with dementia often dislike structured exercise’ [65]. These barriers emphasize the need for enjoyment and accommodation of PA to the preferences of the person with dementia with appropriate activities. This is in line with previous findings stating that retaining flexibility and self-reliance as well as enjoyment and positive experiences with PA are important facilitators and motivators to participate in PA [38,57,66,67,75]. 

New motivators are ‘using preferred or familiar activities’ [53,58], ‘person with dementia chooses PA’[53] and ‘doing activities that are related to everyday life’ [60]. These motivators sharpen the ‘individual tailoring/personalized PA’ [41] by suggesting that specific activities are suited for people with dementia, such as gardening, dog walking, mountaineering or duo-cycling [34]. Providing preferred activities related to everyday life can contribute to the sense of doing meaningful activities, which can increase the well-being and quality of life of people with dementia [76]. Furthermore, having a meaningful purpose for PA encourages people with dementia to participate in PA [38,56,63]. 

### 4.3. New Findings on the Interpersonal Level 

#### 4.3.1. General Support from Informal Care, Family or Health Professional

New barriers are ‘lack of practical and emotional support’ [56], ‘lack of guidance at home’[55] and ‘the person with dementia not living together with the caregiver’ [54]. First, ‘lack of practical and emotional support’ may lead to ‘not having a safe environment on an emotional or practical level’, which was previously identified as a barrier to PA participation [41]. An adequate amount and type of support can create a safe environment for people with dementia. For example, a person with dementia with a strong level of fear of falling may need more encouragement and individual supervision from family and health professionals to feel safe participating in PA [56]. Second, the new barrier ‘lack of guidance at home’ implies that more support for the person with dementia and the caregiver is needed when performing exercises at home [55]. Finally, ‘the person with dementia not living together with the caregiver’ can be a barrier when using a dyadic approach in which both the caregiver and person with dementia participate in PA. If a dyadic approach is used, careful consideration of the role of the caregiver is needed to facilitate PA. In addition, a person with dementia living alone misses potential encouragement and supervision from a partner. In this situation, additional formal support from a professional is needed. 

A new motivator is ‘doing it together with caregiver’ [54] and new facilitators are ‘quality/trained instructors’ [52,55,58,62], ‘creating a positive environment’ [55] and ‘building a trusting relationship between person with dementia, caregiver and trainer’ [50,62]. The motivator and facilitators affirm and extend the previously found facilitator ‘support from (family) caregiver and professional’ [41,62,66] and stress the need to create a safe environment in which the person with dementia feels comfortable and familiar. In this context, it is important that the person with dementia and his or her caregiver can rely on the expertise of the trainer or instructor [77].

#### 4.3.2. Support for Staff in Nursing Homes

New barriers are ‘understaffing’ [58,59], ‘limited time for personalized care and stimulation’ [58] and ‘staff doubts about the PA’ [58]. These barriers were not previously identified by Van Alphen, Hortobágyi and van Heuvelen [40] and Vseteckova et al. [41]. However, Portegijs et al. [78] recently observed that low staff levels and time restraints limit PA in long-time care. Staff doubts may refer to lack of knowledge regarding the benefits of PA and/or how to accomplish these benefits, and lack of confidence that the person with dementia can safely perform PA. Strategies to overcome these barriers, such as appointing staff dedicated to PA and educating or training staff, are limited by low budgets and available personnel [78,79]. 

New facilitators are ‘benefits for staff’ [57,58] and ‘staff prepared to deliver PA’ [58,59]. The facilitator *‘*benefits for staff’ was not previously mentioned and suggests, together with de previously found facilitator ‘experiencing benefits for the caregiver’ [65,80], that people involved in the support of people with dementia (e.g., family caregiver, staff in nursing homes) may experience beneficial effects. An example of such a beneficial effect is improved cooperation during support with daily tasks due to increased physical function of the person with dementia. The new facilitator ‘staff prepared to deliver PA’ can be considered as a generalization of the previously found facilitator training staff to organize the walks’ [41]. Training sessions, consultation and more time can be used to prepare staff to deliver the PA. Education sessions can be used to explain the benefits of the PA to overcome possible doubts about the effectiveness of PA [58].

### 4.4. New Findings on the Community Level

#### 4.4.1. Structural and Organizational (Intervention) Factors: Content of the Intervention

New barriers are the ‘amount of content delivered’ [55] and ‘armrests of chairs’ [60]. These barriers were not previously identified by opinion-based studies. 

New facilitators regarding the content of the intervention are the use of ‘instruction methods’ [53,58,60], ‘intervention accommodation’ [60] and ‘individual supervision’ [60]. People with dementia have difficulty following and remembering instructions. The use of specific ‘instruction methods’, such as individual instructions, short sentences, simple language and demonstration can help to overcome the previously found barrier ‘problems with cognition (attention and memory)’[62,63] and generalizes the previous facilitator ‘slow introduction of new activities’ [64]. The facilitator ‘intervention accommodation’ extends the previously identified facilitator ‘program characteristics’ [60] by suggesting that adjustment to the intervention program may be needed if the intervention program is not suitable for the person with dementia. For example, the intensity or difficulty of exercises can be adjusted to the patient’s capabilities [81]. The facilitator ‘individual supervision’ was not mentioned before and can facilitate ‘the tailoring/personalization of PA’ [41] and effectively motivate people with dementia to carry out the PA [60]. 

#### 4.4.2. Structural and Organizational (Intervention) Factors: Organization of the Intervention

New barriers are ‘costs’ [52], ‘understanding protocol’ [61], ‘adhering to time schedule’ [61], ‘strict timing of walks’ [57] ‘collaboration between experts’ [50] and ‘difficulty transferring dyads to experts’ [50]. These barriers refer mostly to a specific PA intervention. However, some factors are generalizable. For instance, ‘costs’ can be a major barrier for people to participate in PA, while a low-cost PA opportunity facilitates PA participation [50]. Furthermore, ‘adhering to time schedule’ and ‘strict timing of walks’ can hinder PA adherence due to reduced flexibility or competing commitments [52,54,75]. Creating a flexible routine increases PA adherence by allowing people to regularly do the exercises while accounting for other activities and commitments [56]. ‘Collaboration between experts’ and ‘difficulty transferring dyads to experts’ involve the use of experts (general practitioner, physician, physiotherapists, etc.). These barriers add to the previously mentioned barrier ‘obtaining collaboration from care practitioner for medical clearance’ [62] by implying that interdisciplinary collaboration and respectful communication are decisive for successful implementation [50]. 

New facilitators are ‘low-cost PA opportunity’ [50] and ‘support for AAA agencies’ [61]. The facilitator ‘low-cost PA opportunity’ overcomes the barrier ‘costs’ (see above). The facilitator ‘support for AAA’ agencies refers to a specific PA intervention using agencies on aging to implement PA and confirm the new finding that education and support for staff or other involved individuals who implement the PA intervention can facilitate the PA participation of people with dementia (see also Section 4.3.2).

### 4.5. Success Rates of PA Implementations

Objective success rate is generally expressed as rate of adherence (i.e., % of attended sessions/offered sessions). Five studies met the previously set criterion for a successful adherence rate of >75% [82], seven studies did not meet this criterion and one study did not report objective adherence. We found no systematic differences in identified factors affecting PA between objectively more and less successful implementations. Nevertheless, in line with a recent study, we recommend future implementation studies to consistently report adherence rates [83].

### 4.6. Theoretical Framework Integration

In line with Van Alphen, Hortobágyi and van Heuvelen [40], we classified the identified barriers, facilitators and motivators within the social ecological model of McLeroy et al. [43]. However, other frameworks might also be useful in classifying identified factors. For example, The Consolidated Framework for Implementation (CFIR) was developed to ensure that all relevant factors are considered during implementation [37]. The CFIR has similarities with the social ecological model regarding characteristics of the intervention and the individuals involved in the intervention. However, in contrast with the social ecological model, the CFIR consists of separate domains for the organization of the intervention and the process of implementation. In the social ecological model, these factors are combined. The CFIR thus emphasizes the importance of management outside of the intervention. Furthermore, the CFIR is more implementation-focused, while other frameworks related to behavioural change are more person-focused. A new model called the Physical Activity Behaviour Change Theoretical model in dementia (‘PHYT in dementia’) could potentially capture the individual factors needed for behavioural change for people with dementia [84]. Several factors mentioned by the PHYT in dementia, such as self-efficacy, intervention characteristics and the role of the caregiver and health professional correspond to the model of McLeroy et al. [43]. However, the PHYT may give additional insights, since different layers of affecting factors, from global to detailed, are distinguished. Although this model seems promising, it was based on knowledge from people without dementia and needs empirical testing in order to fully explain behavioural change in people with dementia.

### 4.7. Limitations of Underlying Studies and Limitations of This Review

This study has several limitations related to the underlying studies. First, none of the empirical studies differentiated between motivators and facilitators. However, it is important to distinguish them because of differences in impact. Motivators can increase intrinsic motivation and have a stronger and more sustainable impact on PA participation than facilitators, which stimulate extrinsically and should be used repeatedly to maintain impact. Therefore, we differentiated between motivators and facilitators to help inform future studies regarding the design and implementation of PA. 

Second, several empirical studies did not sufficiently describe how the implementation was executed and how the implementation process was evaluated. Consequently, relevant factors affecting PA may be overlooked.

This review has several limitations. First, although we assessed the methodological quality of the studies, we did not explicitly consider the methodological quality to interpret our findings. We used two tools depending on the type of study. The two tools, the CASP (for qualitative studies) and the EPHPP (for quantitative and mixed studies) appeared to not be comparable due to different criteria used (e.g., blinding method and confounders are not specifically assessed with de CASP). Some studies have a low quality according to the EPHPP [60,62]. However, this low quality is related to the study design (quantitative uncontrolled trial), which was critically assessed with the EPHPP. This would not lead to low quality using the CASP. Furthermore, the factors identified by the lower-quality studies of Kruse, Cordes, Schulz and Wollesen [60] and Yu and Kolanowski [62] are comparable to those identified by higher-quality studies. Overall, we assume that the limited quality of some studies has no or a minor impact on our conclusions. 

Second, the studies reported factors directly (e.g., in a table or explicitly mentioned as a factor in the text) or indirectly (mentioned in text, not indicated as a factor). The extraction of indirectly reported factors may be more prone to subjectivity. Furthermore, the fact that only one author (R.W.F.) extracted the data may limit the reliability of the data extraction. Nevertheless, extraction of both directly and indirectly reported factors facilitated a complete overview of barriers, facilitators and motivators to consider when implementing PA. 

Finally, it was not possible to determine which factor is most prominent in a certain PA implementation situation due to the heterogeneity between studies. Comparing studies was not possible due to different types of PA interventions and differences in diagnosis and severity of dementia. We recommend that future studies examine which barriers, facilitators and motivators need to be accounted for during a specific PA implementation situation.

### 4.8. Clinical and Practical Implications

For use in practice, it would be helpful to rank the factors according to importance. However, this kind of ranking is complex for several reasons. First, evaluation of importance depends on the perspective (people with dementia, family caregivers, formal caregivers, researchers). Karssemeijer, De Klijn, Bossers, Olde Rikkert and Van Heuvelen [75] already revealed that people with dementia and their family caregivers tend to judge positive factors not directly affected by dementia (e.g., beneficial health effects) as more important. Formal caregivers, however, attach more importance to barriers related to dementia symptoms (e.g., loss of initiative). Next, importance will depend heavily on the individual’s characteristics and his or her specific environment. So, for one person with dementia, low self-efficacy can be the most important barrier, while for another person, physical problems are the factor most strongly limiting PA. This warrants a personalized approach to PA. 

An important practical implication of our extended overview is that it can facilitate personalized interventions. This overview can be used by (professional) caregivers to tailor PA to the individual’s possibilities and needs. Furthermore, it can help to develop and implement PA intervention strategies for specific dementia subgroups (e.g., dementia patients with specific physical or cognitive limitations or dementia patients with insufficient or ineffective support from family caregivers). Next, our extended list can help researcher further conceptualize personalized PA for people with dementia, i.e., beyond the level of type, intensity and dose of PA [85]. In this context, it can also serve as a framework to develop an online tool to facilitate personalized PA, which can be used to give personalized PA advice to persons with dementia and their caregivers. A final practical implication is that our extended overview illustrates the high complexity of PA implementation. Therefore, it can contribute to the awareness of health care staff and policymakers that effective PA implementation requires a lot of time, money and effort. In addition, it emphasizes the need for specialized professionals to develop implementation strategies and to guide actual implementations so that more people with dementia will experience the beneficial effects of PA.

## 5. Conclusions

This systematic review of implementation studies identified new factors affecting the PA participation of people with dementia, including 21 barriers to PA, and 11 facilitators and four motivators for PA. Previous studies revealed barriers, facilitators and motivators that we did not identify, suggesting that the opinion-based approach and the empirical approach complement each other. New factors are related to the support of people with dementia from informal and formal caregivers, e.g., revealing the importance of a qualified instructor and a trusting relationship. Furthermore, support for staff from an institution or an external party is needed to overcome doubts about PA and prepare staff to implement PA. Lastly, new factors suggested specific recommendations for the content and organization of the PA intervention, such as adapting instructions and the content of intervention. Overall, our extended overview of barriers, facilitators and motivators affecting PA emphasizes the complexity of implementing PA in people with dementia. Our overview can help family and formal caregivers to personalize PA, assist health professionals and researchers in developing PA implementation strategies and actual PA implementation and allow policymakers to free resources to support these implementations.

## Figures and Tables

**Figure 1 behavsci-13-00913-f001:**
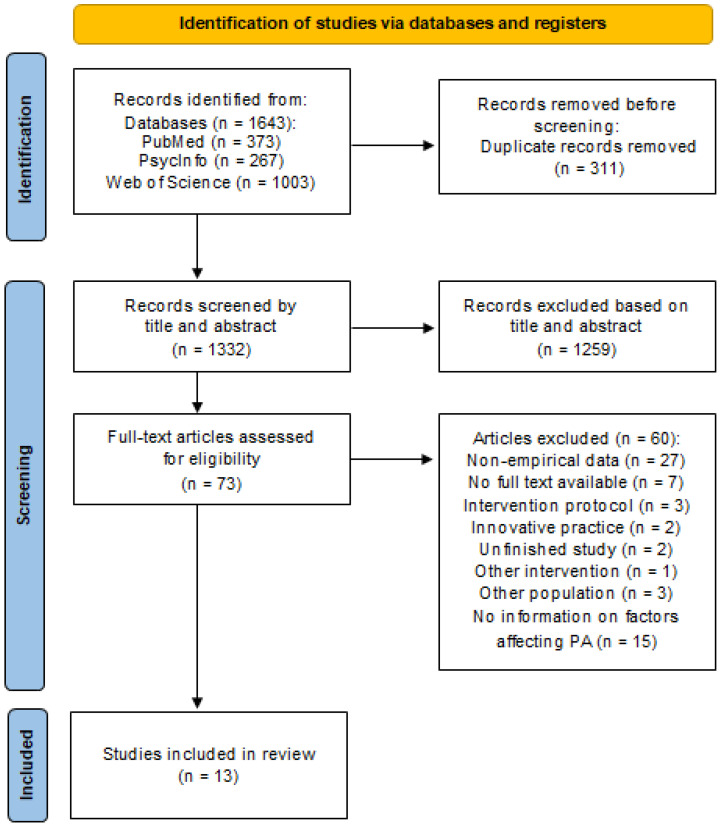
Flowchart of the study selection process (2020 PRISMA Flowchart).

**Table 3 behavsci-13-00913-t003:** Characteristics of included studies.

Reference	Study Design	Participants	Diagnosis of PwD	Age (Mean ± SD) of PwD	% Walking Aid	% Male PwD	Living Situation	Focus on	Type of PA	Duration of PA Intervention (Weeks)	Country
Barrado-Martín, Heward, Polman and Nyman [54]	Qualitative semi-structured interviews; thematic approach	22 PwD 24 caregivers	15 AD 6 mixed 1 Frontal	79.0 ± 6.5	27% WA	55	Community dwelling	Adherence	Tai Chi	20	UK
Barrado-Martín, Heward, Polman and Nyman [55]	Qualitative observations; thematic analyses	10 PwD 10 caregivers	9 AD 1 mixed	78.2 ± 5.4	0% WA	50	Community dwelling	Acceptability	Tai Chi	3–4	UK
Dawson, Judge and Gerhart [53]	Randomized controlled intervention trial	23 PwD	NA MMSE 20.8 ± 5.0	73.9 ± 9.1	NA	43.5	Community dwelling	Facilitation	Functional exercise program (strength and balance)	12	USA
Donkers, van der Veen, Vernooij-Dassen, Nijhuis-van der Sanden and Graff [50]	Qualitative design; thematic analysis	14 PwD 14 caregivers	4 Memory problems 1 MCI 6 AD 3 VaD MMSE 10–24	80.0 ± 9.1	79% WA	42.9	Community dwelling	Delivery	Social Fitness Programme	12	The Netherlands
Hancox, Van Der Wardt, Pollock, Booth, Vedhara and Harwood [56]	Qualitative semi-structured interviews; thematic analysis	20 PwD 19 caregivers	1 MCI 9 AD 4 VaD 4 mixed 2 unknown MMSE 25.1 ± 3.0	76.6 ± 6.6	NA	80	Community dwelling	Adherence	Strength and balance exercises	12–16	UK
Henskens, Nauta, Scherder, Oosterveld and Vrijkotte [58]	Quantitative NCT and qualitative process evaluation	61 PwD	34 AD 7 VaD 6 mixed 14 unknownMMSE 9.8 ± 5.1 (int) and 6.5 ± 5.2 (contr.)	86.5 ± 7.1 84.2 ± 4.7	NA	18.9 29.9	Nursing home	Delivery	Movement-oriented restorative care	52	The Netherlands
Henwood, Neville, Baguley and Beattie [59]	Quantitative NCT	46 PwD	NA	82.4 ± 6.6	NA	40	Nursing home	Delivery	Aquatic exercise program	12	Australia
Kruse, Cordes, Schulz and Wollesen [60]	Quantitative uncontrolled study and qualitative interviews	15 PwD	NA	82 (range 75–90)	NA	36	Nursing home	Feasibility	Multicomponent intervention	16	Germany
MacAndrew, Kolanowski, Fielding, Kerr, McMaster, Wyles and Beattie [57]	Qualitative interviews; thematic analysis	7 PwD	NA	77.0 ± 10.2	NA	43	Nursing home	Feasibility	Walking programme	3	Australia
Sondell, Rosendahl, Gustafson, Lindelöf and Littbrand [51]	Randomized controlled trial	93 PwD	34 AD 36 Vascular 8 Mixed 15 other MMSE 15.4 ± 3.4	84.4 ± 6.2	81% WA	24.7	Nursing home	Applicability	High-intensity functional exercise program	16	Sweden
Tak, van Uffelen, Paw, van Mechelen and Hopman-Rock [52]	Randomized controlled trial and qualitative interviews	134 PwD	134 MCI MMSE 28.4 ± 1.4	74.8 ± 2.9	NA	59	Community dwelling	Adherence	Aerobic exercises	52	The Netherlands
Teri, Logsdon, McCurry, Pike and McGough [61]	Quantitative design and qualitative interviews	255 PwD 20 case manager 10 AAAs	255 AD MMSE 15.6 ± 7.1	81.3 ± 7.7	NA	51	Community residing	Delivery	Multicomponent intervention	6	USA
Yu and Kolanowski [62]	Quantitative uncontrolled trial	2 PwD	2 AD MMSE 17 and 25	75 86	NA	50	Community dwelling	Feasibility	Aerobic exercises	8	USA

Abbreviations: AAAs = Area Agencies on Aging; AD = Alzheimer’s Disease; contr. = control group; int. = intervention group; MCI = Mild Cognitive Impairment; Mixed = mostly Vascular and Alzheimer’s; NA = not available/not announced; NCT = nonrandomized controlled trial; PwD = people with dementia; VaD = Vascular Dementia; WA = walking assistant.

**Table 4 behavsci-13-00913-t004:** Overview of factors influencing PA participation in people with dementia based on implementation- and opinion-based studies, with new factors printed in bold. Within the themes, the factors are ordered by number of studies that identified the factors.

Barriers	Motivators	Facilitators
**Intrapersonal level**		
Physical or Mental Health		
Physical health: Health conditions [41,51,52,54,63,64,65] Fatigue [35,51,63] Mental health: Problems with cognition; attention, memory and confusion [54,56,62,63] Lack of motivation [50,51,62] Disruptive behaviour PwD [35,62] Emotional barriers (fear) [51,63] Depressive symptoms and negative feelings [64] **Low sense of efficacy PwD (or caregiver) [55]** **Lack of confidence at home [55]** Mental well-being [41]	Perceived/experienced physical benefits [35,41,50,52,55,56,57,58,61,65] Improve or maintain physical function/physical well-being [35,63,66] Expectation/belief in possible physical benefits [54,56] Meaningful purpose of PA [56,63] (Awareness of) diagnosis [66]	Physical health: Adapt exercises to physical capabilities/PwD’ needs [51,55,58,60,61,63] Mental health: Strategies to overcome memory problems (memory aids) [35,53,55,56] Mental strategy [35,63] **Behavioural problem solving [61]**
Individual preferences		
**Negative perception/dislike of specific PA [52,55]** **Boredom or lack of enthusiasm [53]** **Not enjoying PA [54]** **Lack of accommodating PA (to preferences) [52]** **Difficulty finding appropriate activities [50]** Dislike of structured exercise [65]	Enjoyment of PA [57,66,67] Positive (past) association/experience with PA [41,56,64] **Using preferred or familiar activities/hobbies [53,58]** Sense of commitment [54] Assist with research [65] **PwD chooses PA [53]** **Activities related to everyday life [60]** Minimize caregiver burden [65] Desire and need to go outdoors [63]	Individual tailoring/personalized PA [41]
**Interpersonal level**		
General support from informal caregiver, family or health professional		
Caregiver factors (burden, doubts) [50,62,64,65] Concerns regarding safety [41,62] **Caregiver and PwD not living together [54]** **Lack of practical and emotional support [56]** **Lack of guidance [55]** Forced/no freedom [64] Relationship dynamics [41] Perceived disruptive behaviours by family [62]	**Doing it together with caregiver [54]** Sense of commitment [65] Relationship with dog [35]	Caregiver/family support/dyadic approach [41,54,55,56,60,62,65,66] Advice, feedback and support experts [50,54,56,58,65] **Quality/trained of instructor/staff [52,55,58,62]** **Trusting relationship trainer, PwD and caregiver [50,62]** Practical strategies [35,63] Educating caregiver and PwD [61,62] **Positive feedback/environment [55]** Attitude of the spouse towards PA [66] Positive experience of spouse [66] Community walking groups [64] Volunteer walking guides [64] Dog walking [64]
Support from staff in nursing homes		
**Understaffing [58,59]** **Limited time for personalized care [58]** **Staff doubts [58]**		**Benefits for staff [57,58]** **Staff prepared to deliver PA [58,59]**
Social identification		
Lack of understanding by other people [64]	Social participation/connectedness [41,50,55] Being with people in the same situation [66,67] Networking [62]	
**Community level**		
Structural and organizational factors		
Limited organized activities [50,62] Practical reasons (time, location) [52,66] Competing commitments (activities, holiday) [54] Obtain collaboration from care practitioner for medical clearance [62] Lack of resources, space/storage/time [67] Logistical barriers: transportation [66] **Strict timing of walks [57]** **Amount of content delivered [55]** **Armrests of chairs [60]** **Costs [52]** **Collaboration between experts [50]** **Difficulty transferring dyads to expert [50]** **Understanding protocol [61]** **Adhering to time schedule [61]**		Development of habit/daily routine [35,54,56,64] **Intervention accommodation [60]** **Instruction methods [53,58,60]** Program characteristics [51,52,65] **Individual supervision [60]** **Low-cost opportunity PA [50]** Tele-health is easy to use [67] Providing transportation [62] Exercise recording [65] Slow introduction of new activities [64] Norms and public health recommendations [63] **Support for AAA agencies [61]**
Physical environment		
Environment (weather, inaccessible, not safe) [41,57,63,64] Difficulty finding the way [35] Being away from home [65]		Avoidance strategies (walking in well-known areas) [63] Safe environment [41]

Opinion-based factors were copied from Van Alphen, Hortobágyi and van Heuvelen [37]. New barriers, facilitators and motivators from this review are presented in bold. Abbreviations: AAAs = Area Agencies on Aging; PwD = people with dementia.

## Data Availability

Not applicable.

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
