# Peer review of "Factors Affecting Physical Activity in People with Dementia: A Systematic Review and Narrative Synthesis"

_behavsci, 2023, doi:10.3390/bs13110913_

Round 1
Reviewer 1 Report
Comments and Suggestions for Authors
Dear authors,
Thank you very much for the submission. This review well elaborated on previous review studies as well as identified new factors affecting physical activity participation of people with dementia, including 22 barriers to physical activity, and 13 facilitators and 4 motivators for PA from different levels including intrapersonal level, interpersonal level, community level, etc. Meanwhile, this review study also differentiated between motivators and facilitators which is potentially very beneficial for future studies regarding the design and implementation of physical activity. However, it would be better to address the importance of different factors in dementia patients as well. I have no further suggestions for this review.
Author Response
Response to reviewer 1
Thank you for your review, nice words and suggestion to improve the paper. If we refer to changes on specific pages/lines in the document, we used Track Changes with All Markup.
“However, it would be better to address the importance of different factors in dementia patients as well.”
Response:
We agree that the topic you addressed, the importance of the factors, is an important issue. However, it is also very complex. How importance of the factors is judged depends on the perspective (e.g., researcher, caregivers, patient) and will also vary per individual (Karssemeijer et al. 2020).
We have now emphasized that in Table 4, the order of the items is based on the number of papers in which the factor is mentioned, which reflect the importance from the perspective of researchers to some extend (page 12, line 300). Furthermore, we discussed the complexity of importance in the discussion section and gave a recommendation for further research (page 20/21, lines 612-625).
Reviewer 2 Report
Comments and Suggestions for Authors
The Authors reviewed systematically the papers which aims to identify new barriers to Physical Activity, and new facilitators and motivators for Physical Activity.
Although the study has the potentiality of being shared with the scientific community, I believe that the manuscript would benefit from a revision with the attempt to better support their experimental setting.
Develop this important parts:
Abstract: include purpose, methods, results, and conclusions of the report
Search strategy: specify the methods used to decide whether a study met the inclusion criteria of the review, including if the reviewers who screened each record and each have worked independently.
Specify the methods used to assess risk of bias in the included studies (with graph)
Provide registration information for the review, including register name and registration number, or state that the review was not registered.
Reviewer 3 Report
Comments and Suggestions for Authors
TITLE AND ABSTRACT
Critical Evaluation:
1. Clarity and Focus: The title is concise and clearly indicates the subject of the study. It effectively communicates the focus on factors influencing physical activity in people with dementia. However, the abstract could provide more specific details about the methodology employed in the study, such as the inclusion criteria, study designs considered, and the period covered in the review.
2. Methodology and Inclusion Criteria: The abstract briefly mentions the systematic searches conducted in Pubmed, PsychInfo, and Web of Science. While it outlines the search terms used, it lacks information about the inclusion and exclusion criteria applied during the study selection process. Providing clarity on these criteria would enhance the transparency and reliability of the review.
3. Key Findings: The abstract presents a summary of the identified factors, including barriers, facilitators, and motivators for PA in people with dementia. The inclusion of new factors not previously addressed in other reviews is a significant strength. However, the abstract does not provide specific examples of these new factors, leaving the reader curious about the nature of these unique findings. Providing illustrative examples or a brief discussion of a few key new factors could enhance the abstract's informativeness.
4. Recommendations for Future Implementations: The abstract concludes by recommending the extended list of barriers, facilitators, and motivators for future PA implementations in people with dementia. While this recommendation is valuable, the abstract could elaborate on how these findings could be practically applied in real-world settings. Providing practical implications or suggestions for healthcare professionals, caregivers, or policymakers would enhance the abstract's applicability.
Suggestions for Improvement:
1. Clarity in Methodology: Provide a concise overview of the inclusion and exclusion criteria, study designs considered, and the period covered in the review. This information is crucial for readers to assess the study's scope and relevance.
2. Illustrative Examples: Include brief examples or anecdotes related to the newly identified factors to provide readers with a better understanding of these unique findings. Concrete examples would make the abstract more engaging and informative.
3. Practical Implications: Elaborate on the practical implications of the study findings. Discuss how the identified factors can inform the development of tailored interventions, support strategies, or policy initiatives. Providing specific recommendations would enhance the abstract's practical utility.
In summary, while the abstract effectively outlines the study's focus and main findings, providing additional details about the methodology, including illustrative examples, and elaborating on practical implications would enhance its clarity, informativeness, and applicability.
INTRODUCTION
The introduction provides a comprehensive overview of the context, significance, and previous research related to physical activity (PA) in people with dementia. It effectively establishes the global prevalence of dementia, outlines the cognitive and physical challenges faced by individuals with dementia, and highlights the importance of PA as a non-pharmacological intervention. The introduction also introduces the concept of barriers, facilitators, and motivators affecting PA participation and identifies the gaps in previous research, setting the stage for the current study's objectives.
Critical Evaluation:
1. Clarity and Context: The introduction is clear and succinct, providing a well-structured background of the topic. It effectively sets the context by presenting relevant statistics and defining key terms. The transition from the global prevalence of dementia to the specific focus on PA is seamless, enhancing the reader's understanding.
2. Literature Review: The introduction integrates previous research findings, citing studies that establish the positive impact of PA on individuals with dementia. The references to previous reviews (Van Alphen et al. [37] and Vseteckova et al. [38]) provide a solid foundation for the study, demonstrating the author's awareness of existing literature.
3. Research Gap and Objectives: The introduction clearly identifies the research gap by pointing out the limitations of previous studies, paving the way for the current study's objectives. The objectives are specific and align with addressing the gaps in the literature, enhancing the study's relevance and importance.
4. Clarity in Terminology: The introduction uses terminology consistently and clearly defines essential terms such as barriers, facilitators, and motivators. This consistency enhances the reader's understanding of the concepts under discussion.
Suggestions for Improvement:
1. Methodological Details: While the introduction outlines the broad methodology (opinion-based approach and empirical approach), it could benefit from brief elaboration on these methods. Providing a concise explanation of how the empirical approach was implemented, the nature of PA interventions studied, and the data collection methods used would enhance the reader's understanding of the study's methodology.
2. Flow and Coherence: While the introduction covers essential aspects, ensuring a seamless flow between sentences and paragraphs could enhance the overall coherence of the text. Ensuring each point logically follows the previous one would make the reading experience smoother for the audience.
METHOD
The methods section describes the search strategy, inclusion/exclusion criteria, data extraction process, categorization of factors, and the methodological quality assessment approach. It provides a clear outline of how the study was conducted and how the data were collected, analyzed, and evaluated.
Critical Evaluation:
1. Clarity and Detail: The methods are described in a clear and concise manner, providing sufficient detail for readers to understand the research process. The inclusion and exclusion criteria are specific and well-defined, ensuring the selection of relevant studies.
2. Search Strategy: The use of multiple databases (Pubmed, PsychInfo, and Web of Science) and a comprehensive search string enhances the comprehensiveness of the literature review. The search strategy is transparent, allowing for replication and verification of the study.
3. Inclusion and Exclusion Criteria: The inclusion and exclusion criteria are appropriately applied, ensuring the selection of studies focusing on PA interventions for people with dementia. The criteria are specific, considering factors such as age, intervention types, and outcome measures.
4. Data Extraction: The data extraction process is clearly described, including the information extracted from each study (study design, participant details, PA intervention information, reported factors, intervention success, and author recommendations). Having one author perform the data extraction ensures consistency.
5. Categorization of Factors: The use of the social-ecological model for categorizing factors provides a theoretical framework, enhancing the organization and understanding of the extracted data. Categorizing factors under themes such as intrapersonal, interpersonal, and community factors aligns with established frameworks in the field.
6. Methodological Quality Assessment: The use of validated tools (CASP for qualitative studies and EPHPP for quantitative/mixed studies) for methodological quality assessment enhances the rigor of the evaluation process. Clear domains for assessment (selection bias, study design, confounders, blinding, data collection method, and withdrawals & dropout) ensure a comprehensive evaluation of study quality.
Suggestions for Improvement:
1. Rationale for Tools: While the use of CASP and EPHPP is appropriate, providing a brief rationale for choosing these specific tools in the context of dementia research would enhance the methodological justification.
2. Inter-Rater Reliability: Although inter-rater reliability is mentioned (e.g., agreement of 98.6%), providing a brief explanation of how discrepancies in study selection or quality assessment were resolved would strengthen the transparency of the process.
3. Integration of Results: While the methods describe the categorization of factors, a brief explanation of how the categorized factors will be integrated into the discussion or analysis would add clarity regarding the study's overall structure.
RESULTS
The results section presents the findings of the study, including the methodological quality assessment of the included studies, the characteristics of the participants and physical activity (PA) interventions, the successes of PA implementation, and the identified barriers, facilitators, and motivators for PA participation in people with dementia.
Critical Evaluation:
1. Methodological Quality Assessment: The results of the methodological quality assessment are clearly presented, indicating the strengths and weaknesses of the included studies. The use of established assessment tools (CASP for qualitative studies and EPHPP for quantitative/mixed studies) enhances the credibility of the evaluation. The specific criteria assessed are outlined, providing transparency in the evaluation process.
2. Study Characteristics: The presentation of study characteristics, including participant demographics and types of PA interventions, is detailed and informative. However, it would be beneficial to include a brief rationale for the selection of specific studies and the diversity of the sample to provide context for readers.
3. Successes of PA Implementation: The objective and subjective measures of PA implementation success are clearly outlined, demonstrating the adherence rates and subjective evaluations of the implemented interventions. Providing a brief comparison or discussion of these success rates in the context of previous literature could add depth to the interpretation of the results.
4. Barriers, Facilitators, and Motivators: The comprehensive list of barriers, facilitators, and motivators for PA participation in people with dementia is well-organized and categorized. The inclusion of previously identified factors and new factors adds depth to the analysis. However, there could be a more explicit discussion of how these factors align with existing theoretical frameworks or models related to PA behavior in the elderly or individuals with cognitive impairments.
Suggestions for Improvement:
Theoretical Framework Integration: Relating the identified barriers, facilitators, and motivators to existing theoretical frameworks (such as health behavior models) would enhance the theoretical grounding of the study. Explaining how the identified factors align with established theories could strengthen the discussion and implications of the findings.
DISCUSSION AND LIMITATIONS
The discussion and conclusions section of the scientific article presents a comprehensive overview of the findings, comparing and contrasting them with previous research. The authors have identified barriers, facilitators, and motivators for physical activity (PA) in people with dementia, emphasizing the importance of these factors in designing effective interventions. Below is a critical evaluation of the text:
Strengths:
1. Comprehensive Review: The discussion provides a detailed analysis of the empirical findings, highlighting both commonalities and new insights. This comprehensive approach enhances the understanding of the factors affecting PA in people with dementia.
2. Integration of Different Levels: The discussion is structured around intrapersonal, interpersonal, and community levels, aligning with the socio-ecological model. This structured approach aids in categorizing and understanding the multifaceted factors influencing PA participation.
3. Acknowledgment of Limitations: The authors transparently acknowledge limitations, including methodological concerns in underlying studies. Addressing these limitations enhances the credibility of the review.
4. Recommendations for Future Research: The discussion provides valuable recommendations for future studies, emphasizing the need for further investigation into specific factors influencing PA implementation in diverse situations.
Areas for Improvement:
1. Clarity in Presentation: While the discussion is comprehensive, the presentation could be more reader-friendly. Breaking down complex sentences and concepts might enhance clarity, ensuring that readers, including those without specialized knowledge, can follow the arguments more easily.
2. Contextualization of Findings: The discussion lacks specific examples or case studies to illustrate the identified factors. Including real-world examples could enhance the practical relevance of the findings and help readers relate to the discussed factors more effectively.
3. Depth of Analysis: While the discussion covers a wide array of factors, a deeper analysis of a few key factors could provide richer insights. For instance, exploring the impact of psychological factors (such as self-efficacy) in greater detail could enhance the discussion's depth.
4. Integration with Existing Literature: While the authors acknowledge the complementarity of opinion-based and implementation studies, a more integrated discussion on how these studies can inform each other would strengthen the argument. Exploring potential frameworks or methodologies that bridge these gaps could add depth to the conclusion.
5. Clarity in Recommendations: The recommendations for future studies are valid, but they could be more specific. Providing concrete suggestions for the methodologies or specific aspects that future research could focus on would enhance the utility of these recommendations.
6. Conclusion Synthesis: The conclusion could synthesize the key findings in a more concise manner, emphasizing the actionable insights for researchers, practitioners, and policymakers. A clear takeaway message summarizing the implications of the study would enhance the overall impact of the conclusion.
In summary, the discussion and conclusions provide a comprehensive overview of the study's findings and their implications. However, enhancing the clarity of presentation, providing specific examples, and delving deeper into a few key factors could further strengthen the discussion section. Additionally, integrating findings with existing literature and providing more specific recommendations for future research would enhance the overall impact of the conclusions.
Author Response
Response to reviewer 3
Thank you for your review and friendly words. We appreciate the time and effort spend on this review and the valuable comments. Please find below our response on the suggestions for improvement point-by-point. If we refer to changes on specific pages/lines in the document, we used Track Changes with All Markup.
TITLE AND ABSTRACT
“1. Clarity in Methodology: Provide a concise overview of the inclusion and exclusion criteria, study designs considered, and the period covered in the review. This information is crucial for readers to assess the study's scope and relevance.”
Response: We added this requested information in the abstract on page 1, lines 15-21. We asked the editor for permission to exceed the word limit of 200 words and she responded that this was no problem.
“2. Illustrative Examples: Include brief examples or anecdotes related to the newly identified factors to provide readers with a better understanding of these unique findings. Concrete examples would make the abstract more engaging and informative.”
Response: We included some concrete examples in the abstract (page 1, lines 24-29).
“3. Practical Implications: Elaborate on the practical implications of the study findings. Discuss how the identified factors can inform the development of tailored interventions, support strategies, or policy initiatives. Providing specific recommendations would enhance the abstract's practical utility.”
Response: We extended the abstract with specific recommendations (page 1, lines 29-32) and in line with this we also extended the discussion (page 20/21, lines 612-644). However, in order to stay within a reasonable word limit, we have to be concise in the abstract.
INTRODUCTION
“1. Methodological Details: While the introduction outlines the broad methodology (opinion-based approach and empirical approach), it could benefit from brief elaboration on these methods. Providing a concise explanation of how the empirical approach was implemented, the nature of PA interventions studied, and the data collection methods used would enhance the reader's understanding of the study's methodology.”
Response: We have now added information in the introduction section with examples regarding the opinion-based approach and the empirical approach, the nature of the PA intervention and the data collection methods (page 2/3, lines 93-95 and 97-105).
“2. Flow and Coherence: While the introduction covers essential aspects, ensuring a seamless flow between sentences and paragraphs could enhance the overall coherence of the text. Ensuring each point logically follows the previous one would make the reading experience smoother for the audience.”
Response: We critically considered our introduction again and made several adaptations to enhance the flow. We especially rephrased the text on page 2, lines 76-83.
METHOD
“1. Rationale for Tools: While the use of CASP and EPHPP is appropriate, providing a brief rationale for choosing these specific tools in the context of dementia research would enhance the methodological justification.”
Response: We have now added some additional argumentation about the relevance of the CASP and EPHPP in the context of dementia (page 4, lines 189-191 and 199-201).
“2. Inter-Rater Reliability: Although inter-rater reliability is mentioned (e.g., agreement of 98.6%), providing a brief explanation of how discrepancies in study selection or quality assessment were resolved would strengthen the transparency of the process.”
Response: We have now added additional information about the study selection process (we distinguish title/abstract and full-text screening with separate agreement rates, page 3, lines 147-153) and the quality assessment (page 4, lines 205-206).
“3. Integration of Results: While the methods describe the categorization of factors, a brief explanation of how the categorized factors will be integrated into the discussion or analysis would add clarity regarding the study's overall structure.”
Response: We have now added information about how we aggregated the categorized factors in the method section (page 4, lines 178-184).
RESULTS
- From critical evaluation: “Study Characteristics: The presentation of study characteristics, including participant demographics and types of PA interventions, is detailed and informative. However, it would be beneficial to include a brief rationale for the selection of specific studies and the diversity of the sample to provide context for readers.”
Response: We added some examples to illustrate the diversity of the sample (page 8, lines 271-280) and examples of PA interventions are reported on page 8/9, lines 286-292. In addition, we added some information on the selection of the studies on page 6, lines 261-264.
- From critical evaluation: “Successes of PA Implementation: The objective and subjective measures of PA implementation success are clearly outlined, demonstrating the adherence rates and subjective evaluations of the implemented interventions. Providing a brief comparison or discussion of these success rates in the context of previous literature could add depth to the interpretation of the results.”
Response: We have added a few sentences in the discussion section regarding the success rates of the studies (page 19, lines 516-525).
- “Theoretical Framework Integration: Relating the identified barriers, facilitators, and motivators to existing theoretical frameworks (such as health behavior models) would enhance the theoretical grounding of the study. Explaining how the identified factors align with established theories could strengthen the discussion and implications of the findings.”
Response: We agree with the reviewer that, next to the social-ecological model, other models may be of relevance. Examples of such models are the implementation-focused Consolidated Framework for Implementation (CFIR) and the recently developed and person-focused Physical Activity Behaviour Change Theoretical model in dementia (‘PHYT in dementia) model . We chose for the social-ecological model as framework to stay in line with prior research of van Alphen et al. and for reasons of clarity for the reader. Nevertheless, we feel that the reviewer addresses an important issue. Therefore, we added a paragraph to the discussion section regarding integrations with the above-mentioned theoretical models (page 19, lines 535-559).
DISCUSSION AND LIMITATIONS
“1. Clarity in Presentation: While the discussion is comprehensive, the presentation could be more reader-friendly. Breaking down complex sentences and concepts might enhance clarity, ensuring that readers, including those without specialized knowledge, can follow the arguments more easily.”
Response: We critically considered our entire discussion section and made several edits to facilitate easy reading (pages 15-21).
“2. Contextualization of Findings: The discussion lacks specific examples or case studies to illustrate the identified factors. Including real-world examples could enhance the practical relevance of the findings and help readers relate to the discussed factors more effectively.”
Response: We have now added several examples to illustrate identified factors (support at home, (page 16, lines 369-373), support to create safe environment (page 17, lines 412-415), contribute to trusting relationship (page 17, lines 430-434) and adjusting the program (page 18, lines 479-482).
“3. Depth of Analysis: While the discussion covers a wide array of factors, a deeper analysis of a few key factors could provide richer insights. For instance, exploring the impact of psychological factors (such as self-efficacy) in greater detail could enhance the discussion's depth.”
Response: We have now elaborated the following factors with more detail: self-efficacy (page 15/16, lines 352-362), behavioral problem solving (page 16, lines 375-383) and meaningful activities (page 16, line 400-404).
“4. Integration with Existing Literature: While the authors acknowledge the complementarity of opinion-based and implementation studies, a more integrated discussion on how these studies can inform each other would strengthen the argument. Exploring potential frameworks or methodologies that bridge these gaps could add depth to the conclusion.”
Response: We combined this point with the point given for the results section (theoretical frame work integration). Therefore, we refer to our response above. Furthermore, we elaborated stronger on the complementarity findings of opinion-based and implementation studies on page 15, lines 327-339. Although we put effort to the section about other frameworks/models, we are not completely sure that we addressed this suggestion completely satisfactory. Other relevant models also apply strongly on empirical findings and therefore do not bridge the (partial) gap between the two approaches.
“5. Clarity in Recommendations: The recommendations for future studies are valid, but they could be more specific. Providing concrete suggestions for the methodologies or specific aspects that future research could focus on would enhance the utility of these recommendations.”
Response: We extended the discussion with concrete clinical and practical implications (page 20/21, lines 612-644). Based on these considerations we extended the conclusion (page 21, lines 652-669) as well as the abstract (page 1, lines 30-33).
“6. Conclusion Synthesis: The conclusion could synthesize the key findings in a more concise manner, emphasizing the actionable insights for researchers, practitioners, and policymakers. A clear takeaway message summarizing the implications of the study would enhance the overall impact of the conclusion.”
Response: As reported in the previous point we extended the conclusion with clinical and practical implications for several stakeholders (page 21, lines 664-669).
Reviewer 4 Report
Comments and Suggestions for Authors
Dementia is characterized by a syndrome manifesting salient features of cognitive impairment. The present study ostensibly conducts a systematic review concerning factors influencing physical activity.
Introduction
While physical activity administered to dementia patients has been reported to facilitate overall cognitive function enhancement, it presents challenges in ameliorating specific cognitive abilities such as memory and attention. Consequently, to validate the legitimacy of the present research, a more meticulous and comprehensive substantiation is requisite, elucidating the efficacy of provided physical activities in mitigating symptoms of dementia.
Inclusion and Exclusion Criteria
While the inclusion criteria are articulated with specificity, the exclusion criteria present a palpable lack of operability. A more definitive and clear exposition of the exclusion criteria is imperative.
Methodological Quality Assessment
In the results of the 'CASP Qualitative Research Checklist,' item 6 is marked as ' can’t tell (?)' in all research instances. A comprehensive explanation regarding these indeterminable statuses is requisite.
Table 2, Table 3
Quantitative studies, namely those by La Rue, et al. [54], La Rue, et al. [55], indicate depreciated quality upon qualitative assessment and, particularly referencing Table 3, appear to be currently underway. This issue seems to emanate from a lack of criteria excluding unfinished studies in the exclusion criteria. Hence, excluding these two studies, predicated upon established exclusion criteria, seems justifiable.
Table 4
The table, elucidating factors impacting participation in physical activity and delineating barriers, motivations, and facilitators, currently suffers from substantial legibility issues from a reader’s perspective. Enhancements in table formatting, among other adjustments, are necessary to augment readability.
4.1. Findings
If there is alignment, even partially, with previous studies, a more detailed exposition is warranted.
4.3. New Findings on the Interpersonal Level
This section proficiently communicates findings related to the physical activity of dementia patients from the study and is perceived as effectively conveyed.
Author Response
Response to reviewer 4
Thank you for your time and effort to review our paper and for your valuables comments and suggestions to improve the paper. Please find below our response point-by-points. If we refer to changes on specific pages/lines in the document, we used Track Changes with All Markup.
Introduction
- “While physical activity administered to dementia patients has been reported to facilitate overall cognitive function enhancement, it presents challenges in ameliorating specific cognitive abilities such as memory and attention. Consequently, to validate the legitimacy of the present research, a more meticulous and comprehensive substantiation is requisite, elucidating the efficacy of provided physical activities in mitigating symptoms of dementia.”
Response: We extended the introduction section. We now distinguished effects of PA on global cognition and on the specific domains executive function on memory. We addressed the effects more critically and related them to specific dementia symptoms (page 2, lines 59-66).
Inclusion and Exclusion Criteria
- “While the inclusion criteria are articulated with specificity, the exclusion criteria present a palpable lack of operability. A more definitive and clear exposition of the exclusion criteria is imperative.”
Response: We agree that the exclusion criteria were reported with insufficient detail. We now described the exclusion criteria more extensively and more critically (page 3, lines 147-153).
Methodological Quality Assessment
- “In the results of the 'CASP Qualitative Research Checklist,' item 6 is marked as ' can’t tell (?)' in all research instances. A comprehensive explanation regarding these indeterminable statuses is requisite.”
Response: We now explained why item 6 is judged with can‘t tell (?) in all papers (page 5, lines 218-225).
Table 2, Table 3
- “Quantitative studies, namely those by La Rue, et al. [54], La Rue, et al. [55], indicate depreciated quality upon qualitative assessment and, particularly referencing Table 3, appear to be currently underway. This issue seems to emanate from a lack of criteria excluding unfinished studies in the exclusion criteria. Hence, excluding these two studies, predicated upon established exclusion criteria, seems justifiable.”
Response: With the more extensive description of the exclusion criteria, we discussed the inclusion of these two studies again. We now decided to exclude these two papers. We adapted the flow chart on page 8 and related text on page 8, lines 269-270. We removed the studies in Table 2 (page 6), Table 3 (page 10-11), Table 4 (page 12-13) and both supplementary files. Since we disagreed in our initial independent screening (and now agree) we also adapted the percentage agreement (page 4, line 148).
Table 4
- “The table, elucidating factors impacting participation in physical activity and delineating barriers, motivations, and facilitators, currently suffers from substantial legibility issues from a reader’s perspective. Enhancements in table formatting, among other adjustments, are necessary to augment readability.”
Response: We formatted this table in accordance with the MDPI guidelines. We asked the editor if it was allowed to deviate from this format. She replied with “we will communicate with you and confirm the format issues at the proofreading stage. This will not be a factor in whether the manuscript will be published, so please do not worry.” Therefore, we made some adaptations in layout (page 12-14), but we are not sure if these adaptations will sustain in a next stage.
4.1. Findings
- “If there is alignment, even partially, with previous studies, a more detailed exposition is warranted.”
Response: We have now discussed the alignment with the previous reviews from Alphen et al. and Vseteckova et al. more thoroughly (page 15, lines 327-340). In additions, we referred to specific additional studies more often, e.g., Karssemeijer et al. 2020 (page 16, line 394; page 18, line 493), Telenius et al. (page 16, line 394), Tierney et al. (page 16, line 403), Portegijs et al. (page 17, line 439) Müllers et al. (page 21, line 634), Sobol et al. (page 16, line 358), Olsen et al (page 16, line 358), Cox et al. (page 16, line 358), Teri et al. (page 16, line 379), Yu et al. (page 16, line 381) and Cohen et al. (page 16, line 381), Long et al. (page 17, line 434).
4.3. New Findings on the Interpersonal Level
“This section proficiently communicates findings related to the physical activity of dementia patients from the study and is perceived as effectively conveyed.”
Response: Thank you for these nice words.